# Antenatal Growth, Gestational Age, Birth, Enteral Feeding, and Blood Citrulline Levels in Very Low Birth Weight Infants

**DOI:** 10.3390/nu16040476

**Published:** 2024-02-07

**Authors:** Midori Obayashi, Sachiko Iwata, Tomoya Okuda, Ichita Mori, Shigeharu Nakane, Yasuko Togawa, Mari Sugimoto, Takao Togawa, Kanji Muramatsu, Shinji Saitoh, Takahiro Sugiura, Osuke Iwata

**Affiliations:** 1Department of Pediatrics and Neonatology, Toyohashi Municipal Hospital, 50 Aza Hakken Nishi, Aotake-cho, Toyohashi 441-8570, Japanwh1te.daem0n.to@gmail.com (T.O.); mori1ichita@gmail.com (I.M.); s.nakane89@gmail.com (S.N.); yasuda-mari@toyohashi-mh.jp (M.S.);; 2Center for Human Development and Family Science, Department of Pediatrics and Neonatology, Nagoya City University Graduate School of Medical Sciences, 1 Kawasumi, Mizuho, Nagoya 467-8601, Japan; s.iwata@med.nagoya-cu.ac.jp (S.I.); ss11@med.nagoya-cu.ac.jp (S.S.)

**Keywords:** birth weight, citrulline, enteral nutrition, gestational age, very low birth weight infant

## Abstract

Early enteral nutrition using reliable biomarkers of intestinal function must be established to improve neurodevelopmental outcomes in very low birth weight infants (VLBWIs). Serum citrulline levels reflect the intestinal function in adults. To elucidate the relationship among antenatal growth, postnatal enteral nutrition, and blood citrulline levels, a retrospective single-center observational study was conducted on 248 VLBWIs born between April 2014 and March 2021. A mixed effect model and post hoc simple slope analysis were used to estimate the correlations between clinical variables and citrulline levels at Early (day 5.1) and Late (day 24.3) postnatal ages. Greater gestational age, birth weight, and amount of enteral nutrition at the time of blood sampling were associated with lower citrulline levels at the Early postnatal age and higher citrulline levels at the Late postnatal age. Provided that Early citrulline levels predominantly reflect the consequence of antenatal citrulline metabolism, it is suggested that fetal growth and maturation are likely to promote citrulline catabolism in utero and its synthesis after birth. With additional insights into the temporal transition point wherein the maturation-dependent balance of citrulline metabolism shifts from catabolism-dominant to synthesis-dominant, citrulline emerges as a potential biomarker for assessing intestinal function and gastrointestinal disorders.

## 1. Introduction

According to a recent report by the World Health Organization, approximately 45% of children under the age of 5 who succumb to mortality are newborns, with 60–80% of these being premature or small for gestational age [1]. Very low birth weight infants (VLBWIs), born less than 1500 g, account for 1.1% of all births in the United Kingdom and display a mortality rate of 17.7% by 1 year of age [2]. A study of Japanese VLBWIs reported that 15.9% of survivors developed moderate to severe neurodevelopmental impairments with developmental quotients below 70 at 3 years of age [3]. VLBWIs encounter challenges in establishing enteral nutrition for various reasons, including immature gut function and unstable intestinal blood supply [4,5]. A study revealed that VLBWIs whose body weight at discharge was less than −2 and −3 standard deviations (SD) compared to the intrauterine growth chart were related with increased odds ratios of 1.43 and 2.93, respectively, for moderate to severe neurodevelopmental impairments at the age of 2 years [6].

Recently, early aggressive parenteral nutrition has been encouraged to prevent postnatal growth restriction after malnutrition in VLBWIs. National Institute for Health and Care Excellence guidelines recommended that intravenous amino acid supplementation for preterm infants should be initiated at 1.5 to 2.0 g/kg/day within the first four days of life, aiming to increase to 3.0 to 4.0 g/kg/day thereafter [7]. High-dose amino acid supplementation up to 3.5 g/kg/day in VLBWIs resulted in improved cognitive-adaptive developmental quotients of 89.5 at 3 years of age compared to 83.1 for low-dose amino acid supplementation up to 1.0 g/kg/day [8]. However, considering that 15% of newborn infants who required parenteral nutrition for or longer than one week developed cholestasis [9] and that catheter related bloodstream infection was reported with the incidence of 3.8–11.3 infections per 1000 catheter days [10], early and safe enteral feeding should be primarily promoted for these vulnerable infants.

A large-scale observational study from the Korean Neonatal Network suggested that early establishment of full enteral feeding before 15 days of life was associated with a reduced risk of developing growth failure at discharge and moderate to severe neurodevelopmental impairments with relative risks of 0.69 and 0.77, respectively, compared to those with delayed establishment of feeding between 16 and 30 days of life [11]. However, a rapid increase in enteral feeding is associated with an elevated risk of developing necrotizing enterocolitis (NEC) [12], the development of which is associated with death in 20–30% of affected VLBWIs [13]. These findings suggest that reliable indicators of intestinal function are required for the early and safe establishment of enteral feeding.

In VLBWIs, gastric residual volumes are routinely assessed to estimate feeding tolerance; however, recent studies have shown that monitoring of gastric residual volumes has little effect on preventing NEC and may increase the time required to establish full enteral feeding [14]. Abdominal radiography findings are also used to ensure safe progression of enteral feeding, although the sensitivities of pneumatosis and pneumoperitoneum to detect NEC remain at 44% and 52%, respectively [15]. Several biochemical markers of NEC have been proposed, including stool calprotectin and blood amyloid A, apolipoprotein C2, and citrulline. Amyloid A is considered an acute inflammatory marker, whereas apolipoprotein C2 is an essential activator of lipoprotein lipase, which shows lower levels in preterm infants with sepsis [16,17]. Elevated amyloid A levels and reduced apolipoprotein C2 levels (197.1 μg/mL and 29.9 μg/mL, respectively) were observed in VLBWIs with sepsis or NEC compared to their healthy peers (32.7 μg/mL and 49.7 μg/mL, respectively) [17], although these biomarkers did not discriminate between NEC and nonabdominal sepsis [13,18]. Stool calprotectin, a protein derived from neutrophils, is a nonspecific biomarker of gut injury [19]. An approximately three-fold increase in stool calprotectin levels was observed in infants with NEC compared to those in their peers [20], although the positive predictive value to identify NEC remains at 0.20; neonates with NEC often do not pass stools, rendering this biomarker difficult to use in VLBWIs [13].

Citrulline is synthesized from glutamine and metabolized to arginine in the intestines of newborn infants [21]. For adult patients with short bowel, citrulline levels less than 20 mmol/L effectively distinguished those with intestinal failure from their peers, with a sensitivity of 92% and a specificity of 90% [22]. Citrulline levels also demonstrated linear correlations with both the fraction of enteral calories and bowel length in infants and young children with short bowel syndrome [23]. Neonatal studies have shown associations between low citrulline levels and the incidence of NEC [4]. However, at present, the physiological and pathological properties of blood citrulline levels in neonates are poorly understood, rendering the prediction of the onset of NEC using citrulline levels still challenging. An important property of blood citrulline levels, that complicate their clinical interpretation, are their dynamic temporal changes before and after birth. Amniotic citrulline levels increase until 30 weeks of gestation and decrease until birth [24]. A steady postnatal increase in citrulline levels, from 19 μmol/L on day 2 to 31 μmol/L on day 28, was observed in preterm infants [4]. It is possible that different primary independent variables during the fetal and neonatal periods contribute to dynamic temporal changes in citrulline levels. With further understanding of these independent variables, citrulline could potentially serve as a useful biomarker for intestinal function and bowel diseases, including NEC.

This study aimed to identify the relationship among antenatal growth, gestational age, postnatal enteral feeding, and blood citrulline levels in VLBWIs.

## 2. Materials and Methods

This retrospective observational study was conducted in a tertiary neonatal intensive care unit at Toyohashi Municipal Hospital, Toyohashi, Aichi, Japan. The study protocol was approved by the local Institutional Ethical Review Board (#660). The requirement for informed parental consent was waived as the study used anonymous data obtained for clinical purposes.

### 2.1. Study Population

Between April 2014 and March 2021, 3088 newborn infants were admitted to our unit. Of these, 253 were VLBWIs, all of whom were enrolled in this study, excluding 5 infants who died before the first assessment of blood citrulline levels (Figure 1).

### 2.2. Blood Sampling, Assay, and Data Collection

Blood citrulline levels were assessed as part of a local mass-screening program for newborn infants together with 19 other biomarkers (Newborn Screening Program at the Aichi Health Promotion Public Interest Foundation) [25] using the flow injection method with a tandem mass spectrometer (LCMS 8040; Shimadzu, Kyoto, Japan). In this unit, we aim to obtain the first mass screening blood sample on day 6 ± 2 according to the patients’ conditions and calendar days on which mass screening is unavailable (mostly weekends or holidays). Blood sampling is repeated approximately every 3–4 weeks until (i) enteral feeding reaches 100 mL/kg/day, (ii) body weight reaches 2000 g, and (iii) findings from the previous mass screening do not suggest reevaluation. Blood samples are obtained by heel lance or venipuncture on a filter paper card, dried, and sent to the mass screening center.

For the current study, citrulline data obtained after five postnatal weeks were not included to minimize sampling bias. Patients’ clinical backgrounds were collected from the electronic medical records, including maternal variables (antenatal glucocorticoid, premature rupture of membranes, hypertensive disorders of pregnancy, and chorioamnionitis), variables at birth (sex, gestational age, birth weight, standard score of birth weight, 1- and 5-min-Apgar scores, mode of delivery, multiple birth, and findings from the arterial cord blood gas analysis at birth), and clinical events and therapeutic options during hospitalization (postnatal age and enteral feeding at blood sampling; incidence of septicemia, intestinal diseases with or without requirements for surgical operation, death, severe intraventricular hemorrhage, and ductus arteriosus requiring pharmacological treatments; the use of antibiotic and intravenous nutrition; postnatal age at the commencement of enteral feeding and achieving full-feeding > 100 mL/kg/day). In this unit, indications for prophylactic treatment of infection and patent ductus arteriosus were not prescribed by fixed gestational ages but were determined based on clinical findings.

### 2.3. Statistical Analysis

Values are shown as medians (interquartile ranges), unless otherwise noted. Citrulline concentrations were normalized using a logarithmic transformation. Univariate analysis was performed to assess the crude dependence of citrulline levels on clinical variables, and the mixed-effect model was used to incorporate repeated sampling from the same individual with the patient identity as a random effect and other clinical variables as fixed effects [26]. To examine the impact of major clinical events on the relationship between patient backgrounds and citrulline levels, the same analysis was repeated in a cohort by excluding newborns, who died within 28 days of life or developed grade III/IV intraventricular hemorrhage, sepsis, or intestinal perforation (findings shown in the Appendix A). For these analyses, independent variables with an incidence of less than 10 were not considered because of a lack of analytical power. Interactions between postnatal age at blood collection and selected clinical variables (gestational age, birth weight, cord blood pH at birth, amount of enteral nutrition at blood sampling, use of antibiotics, necrotizing enterocolitis, and any intestinal disease) and their relationship with citrulline levels were assessed using a mixed effect model. These clinical variables were selected based on our interest in their theoretical interactions with age. For combinations of variables with significant interactions, the post-hoc simple slope analysis was performed for citrulline levels at two representative time points of “Early” and “Late” postnatal ages (one standard deviation below and above the mean of postnatal age, respectively). An additional simple slope analysis was performed to assess the relationship between clinical variables and Early and Late citrulline levels with adjustments for gestational age [27]. The statistical findings were presented without correction for multiple comparisons because of the exploratory nature of this study; however, findings with *p*-values less than but close to 0.05 were interpreted cautiously because of their susceptibility to type-1 errors. All statistical analyses were performed using the SPSS Statistics software ver. 28 (IBM Corp, Armonk, NY, USA).

## 3. Results

### 3.1. Study Population

The final study population comprised 248 VLBWIs (Figure 1) who were born at 29.3 (26.9 to 31.1) weeks of gestation (Table 1). Twenty-eight infants developed some form of intestinal diseases. Fourteen infants subsequently died during initial hospitalization. Blood samples were obtained 2 (1 to 2) times at 15 (6 to 22) days of life, when enteral nutrition commenced in 518 (95.4%) infants at 120 (60 to 140) mL/kg. The citrulline levels in the 543 accumulated blood samples measured 15.6 (10.9 to 19.0) μmol/L.

### 3.2. Crude Dependence of Citrulline Levels on Clinical Variables

Univariable analysis showed that citrulline levels were positively associated with postnatal age (regression coefficient [B], 0.017; 95% confidence interval [CI], 0.014 to 0.020), amount of enteral nutrition at blood sampling (per 10 mL/kg/day; B, 0.019; 95% CI, 0.013 to 0.025), and the incidence of grade III/IV intraventricular hemorrhage (B, 0.281; 95% CI, 0.103 to 0.460), and negatively associated with multiple birth (B, −0.106; 95% CI, −0.197 to −0.014), cord blood pH at birth (per 0.1 pH change; B, −0.047; 95% CI, −0.083 to −0.011), BE at birth (B, −0.013; 95% CI, −0.023 to −0.003), and postnatal antibiotics (B, −0.103; 95% CI, −0.196 to −0.010) (Table 2; See Appendix A for findings from an additional analysis in a subcohort, excluding newborns who died within 28 days of life or developed severe intraventricular hemorrhage, sepsis, or intestinal perforation).

### 3.3. Dependence of Citrulline Levels on Postnatal Age, Clinical Variables, and Their Interactions

Significant interactions with postnatal age were observed for gestational age (B, 0.004; 95% CI, 0.003 to 0.004), birth weight (per 100 g; B, 0.004; 95% CI, 0.003 to 0.005), amount of enteral nutrition (per 10 mL/kg/day; B, 0.003; 95% CI, 0.003 to 0.004), postnatal antibiotics (B, −0.009; 95% CI, −0.016 to −0.002), NEC (B, −0.031; 95% CI, −0.047 to −0.015), and any intestinal disease (B, −0.022; 95% CI, −0.032 to −0.012) (see Table 3 for the main effect of each independent variable).

### 3.4. Independent Variables of Citrulline Levels during Early and Late Postnatal Ages

A simple slope analysis was performed to estimate the relationship between independent variables and citrulline levels at Early (day 5.1) and Late (day 24.3) postnatal ages, which were defined by the mean blood sampling date (14.7) and its standard deviation (9.6). During the Early postnatal age, gestational age (B, −0.025; 95% CI, −0.040 to −0.011), birth weight (per 100 g; B, −0.030; 95% CI, −0.046 to −0.015), and amount of enteral nutrition at blood sampling (per 10 mL/kg/day; B, −0.021; 95% CI, −0.028 to −0.014) were negatively associated with citrulline levels (Table 4; Figure 2).

At the Late postnatal age, gestational age (B, 0.042; 95% CI, 0.029 to 0.056), body weight (per 100 g; B, 0.047; 95% CI, 0.031 to 0.062), and amount of enteral nutrition (per 10 mL/kg/day; B, 0.040; 95% CI, 0.030 to 0.050) were positively associated with citrulline levels, whereas postnatal antibiotics (B, −0.185; 95% CI, −0.295 to −0.075), NEC (B, −0.488; 95% CI, −0.728 to −0.247), and any intestinal disease (B, −0.389; 95% CI, −0.548 to −0.230) were negatively associated with citrulline levels.

An additional analysis performed to assess the relationship between clinical variables and Early and Late citrulline levels after adjusting for gestational age revealed that birth weight (per 100 g; B, −0.038; 95% CI, −0.060 to −0.016) and amount of enteral nutrition (per 10 mL/kg/day; B, −0.022; 95% CI, −0.030 to −0.015) were negatively associated with citrulline levels at the Early postnatal age (Table 5). At the Late postnatal age, birth weight (per 100 g; B, 0.039; 95% CI, 0.016 to 0.061) and amount of enteral nutrition (per 10 mL/kg/day; B, 0.038; 95% CI, 0.027 to 0.049) were positively associated with higher citrulline levels, whereas postnatal antibiotics (B, −0.165; 95% CI, −0.280 to −0.050), NEC (B, −0.453; 95% CI, −0.692 to −0.213), and any intestinal disease (B, −0.370; 95% CI, −0.527 to −0.213) were negatively associated with citrulline levels.

## 4. Discussion

Consistent with previous findings in newborns, we confirmed that citrulline levels in VLBWIs increased with postnatal age. When interactions with postnatal age were incorporated into the analysis, greater gestational age, larger birth weight, and larger enteral nutrition at the time of blood sampling were associated with lower citrulline levels within the first week of life and higher citrulline levels after a few weeks of life. These findings suggest that organ maturation accelerates citrulline consumption in utero and synthesis after birth, leading to dynamic and paradoxical temporal changes in citrulline levels.

### 4.1. Gestational and Postnatal Age and Citrulline Levels

Previous studies have reported that citrulline levels in free amniotic fluid at term-equivalent periods are approximately 30% of those during the first trimester [28,29,30], the temporal reduction of which was explained by the maturation-dependent upregulation of arginine synthesis from citrulline towards the end of pregnancy [24]. In contrast, Ioannou et al. showed that the plasma citrulline levels in preterm infants increased with postnatal age. Based on a significant linear relationship between the amount of enteral nutrition and plasma citrulline levels, it was speculated that enteral nutrition is likely to upregulate the intestinal citrulline synthesis after birth [4]. Another study on newborn infants and young children with short bowel syndrome showed that citrulline levels were associated with tolerance of enteral nutrition [23]. Given that citrulline is actively catabolized and synthesized in the small intestine [21] and that the length of the intestine grows approximately 2-fold from 19 to 27 weeks of gestation [31], a greater gestational age and subsequently longer intestine length may simultaneously contribute to greater citrulline consumption in utero and its synthesis after birth. The current study confirmed a linear relationship between postnatal age and blood citrulline levels in VLBWIs. Furthermore, greater gestational age was associated with lower citrulline levels at the Early postnatal age and higher citrulline levels at the Late postnatal age. Considering the time required for VLBWIs to establish full enteral feeding, the Early blood citrulline levels are likely to represent those in fetal circulation, supporting the hypothesis that maturation-dependent upregulation of citrulline consumption in utero and its synthesis after birth are responsible for the paradoxical and dynamic temporal changes in blood citrulline levels after birth.

### 4.2. Body Size, Enteral Nutrition, and Citrulline Levels

In the current study, the effects of birth weight and enteral nutrition on blood citrulline levels were similar to those at gestational age. Initially, we hypothesized that the temporally paradoxical relationships between birth weight, enteral nutrition, and citrulline levels are indirect, likely caused by the close association between gestational age and citrulline levels. However, these relationships were observed even after adjusting for gestational age. Several explanations are possible. First, while gestational is considered as the primary independent variable for organ maturation of newborn infants, other variables, such as maternal, nutritional, and genetic factors, are also crucial [32,33]. These variables might also influence citrulline metabolism. Second, the influence of maturation may not be sufficiently eliminated when gestational age and citrulline levels are nonlinearly associated. Although we found that a greater amount of enteral nutrition at the Early postnatal age was associated with lower citrulline levels, the assumed timing of blood sampling (day 5.1) may be too early to observe the consequences of altered citrulline metabolism, when full enteral feeding is not achieved at that time in most VLBWIs. It is also possible that birth weight and enteral nutrition have direct physiological and pathological effects on citrulline metabolism. Given that both gestational age and birth weight are closely related to intestinal length [31], longer intestines in heavier infants may contribute to higher antenatal consumption and postnatal synthesis of citrulline, independent of other confounders.

### 4.3. Other Independent Variables of Citrulline Levels

In addition to gestational age and birth weight, we found that postnatal antibiotic use and the incidence of NEC and other intestinal diseases were associated with low citrulline levels at the Late time point. Although these clinical conditions may represent the immature state of infants, these relationships were observed even after adjusting for gestational age. It was also interesting that the use of postnatal antibiotics and the incidence of NEC and other intestinal diseases were associated with citrulline levels at the Late, but not Early, postnatal age. Low citrulline levels may be a consequence of delayed enteral nutrition due to the progression of NEC and other intestinal diseases. It is also possible to speculate that the early prescription of antibiotics is deleterious to the acquisition of mature intestinal function via its negative effect on the intestinal flora [34,35]. Further studies with larger populations of VLBWIs are required to elucidate the causal relationships between antibiotics, NEC, intestinal diseases, and Early and Late blood citrulline levels.

### 4.4. Clinical Implication

Based on reports from our group and others, we propose a hypothetical relationship between clinical variables and citrulline levels before and after birth in VLBWIs (Figure 3). The greater maturation state of VLBWIs, as represented by gestational age and birth weight, may lead to upregulated citrulline consumption in utero, resulting in lower citrulline levels within the first week of life. In contrast, a more mature state may lead to greater intestinal tolerance to enteral feeding and upregulate citrulline synthesis, leading to higher citrulline levels after a few weeks of life. Our results suggest that low blood citrulline levels within the first week of life and high citrulline levels after a few weeks of life are suggestive of successful enteral nutrition, possibly supporting the idea that blood citrulline levels may serve as a clinical indicator of residual intestinal function in preterm infants. For VLBWIs whose enteral nutrition is not established after a few weeks of life, high blood citrulline levels might suggest future tolerance of enteral nutrition.

Various biomarkers, including blood amyloid A, apolipoprotein C2, and stool calprotectin, have been suggested as potential indicators for predicting intestinal diseases in neonates [16,17,18,19,20]. However, in contrast to these biomarkers, which respond to inflammation or tissue injury, blood citrulline is more likely to mirror the actual intestinal function [21]. Therefore, blood citrulline levels might be better suited to support tailored enteral nutrition in high-risk neonates. However, to utilize this biomarker as a predictor of successful enteral nutrition in the earlier stages, future studies must identify the threshold citrulline levels to discriminate clinically pathological intestinal functions and the exact timing when the correlation between intestinal function and citrulline levels changes from negative to positive.

### 4.5. Limitations

Before translating our findings into clinical practice, several limitations should be noted. First, assessment of the relationship between citrulline levels and the development of intestinal disorders, such as NEC, meconium ileus, and intestinal perforation, was beyond the scope of our study due to the limited size of the study population. A multicenter study involving many VLBWIs is required to elucidate the relationship between early and late blood citrulline levels and morbidity.

Second, our study used data obtained from a local newborn mass screening program. Subsequently, the measurements of citrulline levels from blood spots were less precisely defined than the standard plasma citrulline assessment, although this level was sufficient to explore the independent variables of blood citrulline levels in a medium-sized cohort of VLBWIs. The timing of blood sampling was uniform and suboptimal for capturing the dynamic postnatal changes in citrulline levels in relation to clinical backgrounds. The number of repeated samplings from the same infant was also limited. However, the timing of the blood sampling was minimally affected by the infant’s clinical condition. In addition, the influence of potential biases from individual clinical backgrounds was eliminated using a mixed effect model, which incorporates repeated sampling from the same infants.

## 5. Conclusions

Blood citrulline levels in VLBWIs increase with postnatal age. Greater gestational age, higher birth weight, and a greater amount of enteral nutrition at the time of blood sampling were associated with lower citrulline levels within the first week of life and higher citrulline levels after a few weeks of life. These findings suggest that fetal maturation and growth are likely to promote citrulline catabolism in utero and its synthesis after birth. High citrulline levels shortly after birth and low citrulline levels after the first week of life may serve as indicators for neonates requiring special attention in enteral feeding. However, blood citrulline levels might not be readily used as a biomarker to predict the residual capacity of intestinal function due to the temporally paradoxical relationship with the amount of enteral nutrition tolerated at the time of blood sampling. Future studies should elucidate (i) the exact timing wherein the maturation-dependent balance of citrulline metabolism shifts from catabolism-dominant to synthesis-dominant, and (ii) whether successful enteral nutrition can be predicted by Early low and Late high citrulline levels by obtaining multiple blood samples within the first two weeks of life in a greater number of newborn infants.

## Figures and Tables

**Figure 1 nutrients-16-00476-f001:**
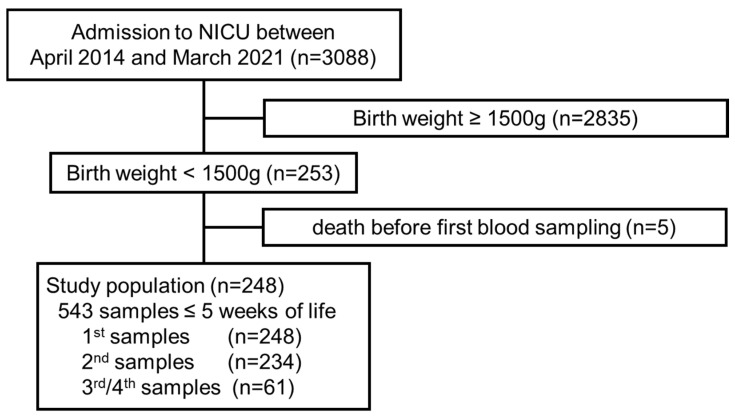
Study population. Flow diagram of the study population and blood samples assessed.

**Figure 2 nutrients-16-00476-f002:**
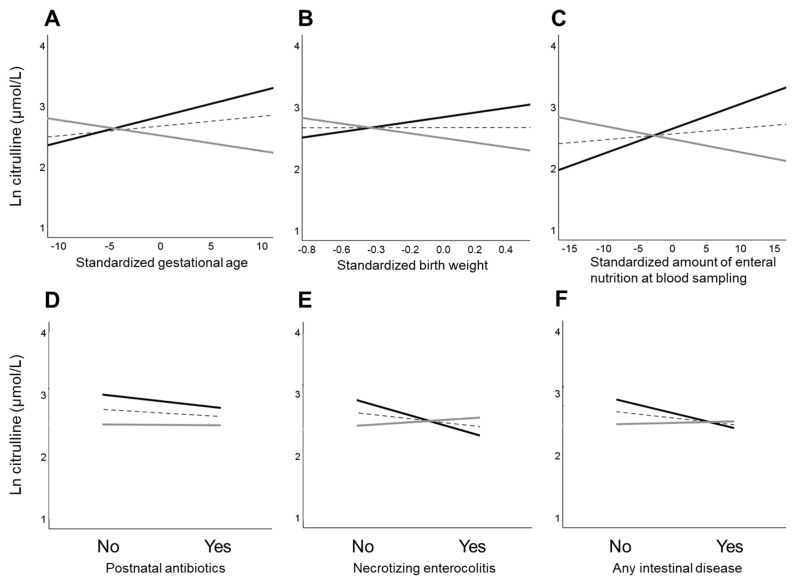
Relationships between clinical variables and citrulline levels at Early and Late postnatal ages. Relationships between gestational age (**A**), birth weight (**B**), amount of enteral nutrition at blood sampling (**C**), use of postnatal antibiotics (**D**), incidence of necrotizing enterocolitis (**E**), any intestinal diseases (**F**), and citrulline levels at Early (one standard deviation below the mean) and Late (one standard deviation above the mean) postnatal ages. Regression lines between clinical variables and blood citrulline levels are shown for Early (solid gray line), Late (solid black line), and mean (dotted line) postnatal ages using simple slope analysis. Continuous variables (gestational age, birth weight, and the amount of enteral nutrition at the time of blood sampling) were mean centered for convenience.

**Figure 3 nutrients-16-00476-f003:**
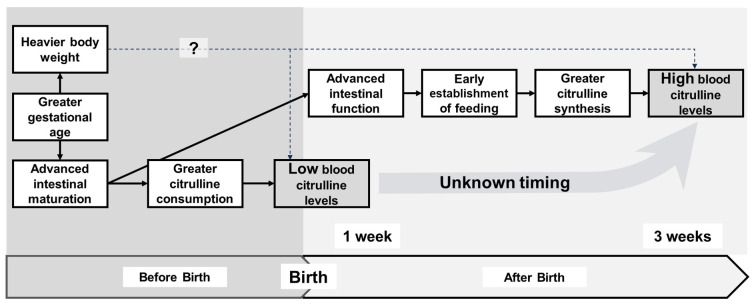
Directional acyclic graph depicting the estimated relationships between clinical variables and citrulline levels. Hypothetical relationships between clinical variables and citrulline levels before and after the birth of very low birth weight infants. A greater maturational state of infants, represented by gestational age and birth weight, may lead to upregulated citrulline consumption in utero, leading to lower citrulline levels within the first week of life. In contrast, a greater maturational state may lead to greater intestinal tolerance to enteral feeding and upregulated citrulline synthesis, leading to higher citrulline levels after a few weeks of life.

**Table 1 nutrients-16-00476-t001:** Clinical backgrounds of the study population.

Variables	n = 248
Maternal variables	
Antenatal glucocorticoid	155 (63%)
Premature rupture of membranes	72 (29%)
Hypertensive disorders of pregnancy	47 (19%)
Chorioamnionitis	16 (6.5%)
Variables at birth	
Male sex	129 (52%)
Gestational age (week)	29.3 (26.9 to 31.1)
Birth weight (g)	1158 (840 to 1338)
Standard score of birth weight	−0.90 (−1.86 to −0.01)
Apgar score (1 min)	6 (3 to 8)
Apgar score (5 min)	8 (7 to 9)
Cesarean delivery	189 (76%)
Multiple birth	46 (19%)
Cord blood pH at birth	7.31 (7.25 to 7.38)
Base Excess at birth (mEq/L)	−2.95 (−4.90 to −0.63)
Clinical events during hospitalization	
Parenteral nutrition	213 (86%)
Antibiotics	202 (82%)
Patent ductus arteriosus requiring pharmacotherapy	60 (24%)
Grade III/IV intraventricular hemorrhage	12 (4.8%)
Septicemia	36 (15%)
Intestinal diseases	
Necrotizing enterocolitis	10 (4.0%)
Intestinal perforation	6 (2.4%)
Meconium ileus	1 (0.4%)
Other intestinal diseases	11 (4.4%)
Requiring surgical intervention	3 (1.2%)
Any of above	28 (11.3%)
Postnatal age at full enteral feeding (day)	9 (7 to 14)
Establishment of enteral nutrition	
<1 week	109 (44%)
<2 weeks	74 (30%)
<3 weeks	37 (15%)
≥3 weeks	28 (11%)
Death	14 (5.6%)

Values are shown as numbers (%) or medians (95% confidence interval).

**Table 2 nutrients-16-00476-t002:** Crude dependence of citrulline levels on clinical variables.

Variables		Regression Coefficient			*p*
		Mean	95% CI		
			Lower	Upper	
Maternal variables					
Antenatal glucocorticoid		−0.032	−0.108	0.043	0.396
Premature rupture of membranes		−0.010	−0.090	0.069	0.796
Hypertensive disorders of pregnancy		−0.063	−0.157	0.031	0.187
Chorioamnionitis		−0.041	−0.146	0.063	0.436
Variables at birth					
Male sex		−0.032	−0.105	0.041	0.384
Gestational age (week)		0.008	−0.004	0.020	0.191
Birth weight (per 100 g)		0.006	−0.006	0.019	0.327
Standard score of birth weight		−0.005	−0.033	0.024	0.742
Apgar score (1 min)		−0.003	−0.018	0.012	0.716
Apgar score (5 min)		−0.005	−0.026	0.016	0.672
Cesarean delivery		0.052	−0.032	0.136	0.227
Multiple birth		−0.106	−0.197	−0.014	0.024
Cord blood pH at birth (per 0.1)		−0.047	−0.083	−0.011	0.010
Base Excess at birth (mEq/L)		−0.013	−0.023	−0.003	0.012
Clinical events during hospitalization					
Parenteral nutrition		−0.022	−0.128	0.084	0.679
Antibiotics		−0.103	−0.196	−0.010	0.029
Patent ductus arteriosus requiring pharmacotherapy		−0.080	−0.165	0.004	0.063
Grade Ⅲ/Ⅳ intraventricular hemorrhage		0.281	0.103	0.460	0.002
Septicemia		0.048	−0.057	0.153	0.369
Necrotizing enterocolitis		−0.157	−0.348	0.033	0.105
Any intestinal disease		−0.156	−0.273	−0.039	0.009
Establishment of enteral nutrition	<1 week	0.139	0.012	0.266	0.032
	<2 weeks	0.096	−0.036	0.228	0.155
	<3 weeks	0.212	0.065	0.360	0.005
	≥3 weeks	Reference			
Postnatal age at blood sampling (day)		0.017	0.014	0.020	<0.001
Amount of enteral nutrition at blood sampling (per 10 mL/kg)		0.019	0.013	0.025	<0.001
Death		0.173	0.000	0.346	0.050

Values are from the mixed effect model. Abbreviation: CI, confidence interval.

**Table 3 nutrients-16-00476-t003:** Dependence of citrulline levels on postnatal age, clinical variables, and their interactions.

Variables	Regression Coefficient			*p*
	Mean	95% CI		
		Lower	Upper	
Variables at birth				
Gestational age (week)	0.008	−0.003	0.020	0.150
Postnatal age (day)	0.016	0.014	0.019	<0.001
Gestational age × Postnatal age	0.004	0.003	0.004	<0.001
Birth weight (per 100 g)	0.008	−0.004	0.021	0.201
Postnatal age (day)	0.016	0.013	0.019	<0.001
Birth weight × Postnatal age	0.004	0.003	0.005	<0.001
pH at birth (per 0.1)	−0.050	−0.085	−0.014	0.006
Postnatal age (day)	0.017	0.014	0.020	<0.001
pH at birth × Postnatal age	0.002	−0.001	0.005	0.136
Postnatal variables				
Amount of enteral nutrition at blood sampling (per 10 mL/kg)	0.009	0.003	0.016	0.006
Postnatal age (day)	0.009	0.005	0.013	<0.001
Amount of enteral nutrition × Postnatal age	0.003	0.003	0.004	<0.001
Postnatal antibiotics	−0.099	−0.191	−0.006	0.036
Postnatal age (day)	0.024	0.018	0.030	<0.001
Postnatal antibiotics × Postnatal age	−0.009	−0.016	−0.002	0.010
Necrotizing enterocolitis	−0.189	−0.378	−0.001	0.049
Postnatal age (day)	−0.013	−0.028	0.002	0.100
Necrotizing enterocolitis × Postnatal age	−0.031	−0.047	−0.015	<0.001
Any intestinal disease	−0.176	−0.293	−0.059	0.003
Postnatal age (day)	0.019	0.016	0.022	<0.001
Any of Intestinal disease × Postnatal age	−0.022	−0.032	−0.012	<0.001

Values are from the mixed effect model. Abbreviation: CI, confidence interval.

**Table 4 nutrients-16-00476-t004:** Independent variables of citrulline levels during the Early and Late postnatal ages.

Variables	Early (Day 5.1)				Late (Day 24.3)			
	Mean	95% CI		*p*	Mean	95% CI		*p*
		Lower	Upper			Lower	Upper	
Gestational age (week)	−0.025	−0.040	−0.011	0.011	0.042	0.029	0.056	<0.001
Birth weight (per 100 g)	−0.030	−0.046	−0.015	<0.001	0.047	0.031	0.062	<0.001
Amount of enteral nutrition at blood sampling (per 10 mL/kg)	−0.021	−0.028	−0.014	<0.001	0.040	0.030	0.050	<0.001
Postnatal antibiotics	−0.012	−0.128	0.104	0.843	−0.185	−0.295	−0.075	0.001
Necrotizing enterocolitis	0.109	−0.131	0.350	0.373	−0.488	−0.728	−0.247	<0.001
Any intestinal disease	0.037	−0.111	0.185	0.625	−0.389	−0.548	−0.230	<0.001

Values are from the simple slope analysis. Abbreviation: CI, confidence interval.

**Table 5 nutrients-16-00476-t005:** Independent variables of citrulline levels during the Early and Late postnatal ages with adjustment for gestational age.

Variables	Early (Day 5.1)				Late (Day 24.3)			
	Mean	95% CI		*p*	Mean	95% CI		*p*
		Lower	Upper			Lower	Upper	
Birth weight (per 100 g)	−0.038	−0.060	−0.016	0.001	0.039	0.016	0.061	0.001
Amount of enteral nutrition at blood sampling (per 10 mL/kg)	−0.022	−0.030	−0.015	<0.001	0.038	0.027	0.049	<0.001
Postnatal antibiotics	0.007	−0.116	0.129	0.913	−0.165	−0.280	−0.050	0.005
Necrotizing enterocolitis	0.142	−0.106	0.390	0.262	−0.453	−0.692	−0.213	<0.001
Any intestinal disease	0.053	−0.099	0.201	0.492	−0.370	−0.527	−0.213	<0.001

Values are from the simple slope analysis with adjustment for gestational age. Abbreviation: CI, confidence interval.

## Data Availability

The data presented in this study are available on request from the corresponding author. The data are not publicly available due to ethical restrictions.

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
