# Peer review of "Antenatal Growth, Gestational Age, Birth, Enteral Feeding, and Blood Citrulline Levels in Very Low Birth Weight Infants"

_nutrients, 2024, doi:10.3390/nu16040476_

Round 1
Reviewer 1 Report
Comments and Suggestions for Authors
This well-written retrospective observational study aimed to explore the associations between prenatal growth, postnatal enteral nutrition, and blood citrulline levels in very low birth weight infants. The study findings suggest that fetal growth and maturation likely promote both intrauterine citrulline breakdown and postnatal synthesis, providing evidence for the transition of citrulline metabolism balance before and after birth. The analytical methods employed in the study were reasonable, and the description and discussion of the results were accurate and comprehensive.
I have a few suggestions to further improve the article:
Provide a reference for the source of the blood sampling criteria. If the sampling criteria align with previous studies, it would be beneficial to cite them as references.
Clarify which variables were considered fixed effects and which were considered random effects in the mixed effect model section.
The study results indicate a potential association between clinical events during hospitalization and outcome measures. It would be helpful to adjust for clinical events during hospitalization in the analysis or conduct sensitivity analyses by excluding individuals who experienced clinical events during hospitalization to enhance the reliability of the results.
The authors mentioned that the assumed timing of blood sampling might be too early to observe changes in citrulline metabolism. It would be advisable to mention this limitation in the limitations section.
Author Response
Comments by the reviewers and author rebuttals:
Reviewer 1
Reviewer comment:
This well-written retrospective observational study aimed to explore the associations between prenatal growth, postnatal enteral nutrition, and blood citrulline levels in very low birth weight infants. The study findings suggest that fetal growth and maturation likely promote both intrauterine citrulline breakdown and postnatal synthesis, providing evidence for the transition of citrulline metabolism balance before and after birth. The analytical methods employed in the study were reasonable, and the description and discussion of the results were accurate and comprehensive.
- Response to the reviewer:
We appreciate the valuable suggestions provided by the reviewer from an expert point of view, which helped clarify and improve the quality of our manuscript. Here is a point-by-point response to your comments.
Reviewer comment:
I have a few suggestions to further improve the article:
Provide a reference for the source of the blood sampling criteria. If the sampling criteria align with previous studies, it would be beneficial to cite them as references.
- Response to the reviewer: We agree to the reviewer that a reference dealing with the analysis using the dried blood spot for newborn mass screening should be cited. The revised reference list includes the following reference: Nakano et al. Development of tandem mass spectrometry-based creatinine measurement using dried blood spot for newborn mass screening. Pediatr Res. 2017;82(2):237-43. We thank the reviewer for the suggestion.
Reviewer comment:
Clarify which variables were considered fixed effects and which were considered random effects in the mixed effect model section.
- Response to the reviewer: We agree that sufficient information should be provided to understand the structure of statistical models. We used the patient identity as a random effect and other clinical variables as fixed effects. The Data analysis section has been revised with additional information. We thank the reviewer for the comment.
Reviewer comment:
The study results indicate a potential association between clinical events during hospitalization and outcome measures. It would be helpful to adjust for clinical events during hospitalization in the analysis or conduct sensitivity analyses by excluding individuals who experienced clinical events during hospitalization to enhance the reliability of the results.
- Response to the reviewer: The reviewer is correct that patients who experienced major clinical events, especially those inducing systemic inflammation, may significantly alter the relationship between the patient backgrounds and citrulline levels. We have performed an additional analysis by excluding newborns with severe intraventricular hemorrhage, sepsis, and intestinal perforation. Subsequently, most relationships (other than those close to the threshold of significance) were preserved. The findings of this analysis are now presented in the Appendix. The authors thank the reviewer for the comment.
Reviewer comment:
The authors mentioned that the assumed timing of blood sampling might be too early to observe changes in citrulline metabolism. It would be advisable to mention this limitation in the limitations section.
- Response to the reviewer: We agree to the reviewer that the limitation regarding the timing of blood sampling should be mentioned in the Limitation section. We have revised the manuscript so that readers can evaluate the influence of the timing to the relationship between clinical variables and citrulline levels. The authors appreciate the reviewer’s suggestion.

Reviewer 2 Report
Comments and Suggestions for Authors
The title of this article is “Antenatal growth, gestational age, birth, enteral feeding, and blood citrulline levels in very low birth weight infants”. This is an interesting topic. However, there are still some areas of the article that need to be revised:
1. In the introduction section of this article, the authors mainly describe the global probability of occurrence of "very low birth weight infants (VLBWIs)" and the progress of research on their markers. However, there are relatively few studies on blood citrulline levels, and it is suggested that more be added.
2. What are the advantages of blood citrulline levels over previous markers used in other studies.
3. In the CONCLUSION it is possible to look forward not only from the future direction of the research, but it is recommended to add an outlook on the future clinical applications of this research.
4. Journal abbreviations should be italicized in references. Also, journals with partial abbreviations that are the same as the full title should not be followed by ".". For example, "Gastroenterology" on line 424.
Comments on the Quality of English LanguagePlease check the English expression of the article again, remove some unnecessary and repetitive words to make the article more concise, and also check the English tense.
Author Response
Comments by the reviewers and author rebuttals:
Reviewer 2
Reviewer comment:
The title of this article is “Antenatal growth, gestational age, birth, enteral feeding, and blood citrulline levels in very low birth weight infants”. This is an interesting topic. However, there are still some areas of the article that need to be revised:
- Response to the reviewer: We are grateful to the reviewer for the constructive criticisms. The manuscript has been revised in light of the suggestions provided by the reviewer. Please find the point-by-point response to the reviewer’s comments.
Reviewer comment:
- In the introduction section of this article, the authors mainly describe the global probability of occurrence of "very low birth weight infants (VLBWIs)" and the progress of research on their markers. However, there are relatively few studies on blood citrulline levels, and it is suggested that more be added.
- Response to the reviewer: The authors agree that the Introduction section need to address the lack of studies on citrulline levels in newborn infants. We have emphasized the point in the revised Introduction section.
Reviewer comment: 2. What are the advantages of blood citrulline levels over previous markers used in other studies.
- Response to the reviewer: The authors appreciate the reviewer’s suggestion that the potential advantage of citrulline monitoring over other biomarkers needs to be clarified. Because citrulline is synthesized in the small intestine, blood citrulline levels are likely to reflect the total amount of functioning intestine. Considering that citrulline levels are monitored in mass screening tests, with further clinical evidence, citrulline-based assessment/prediction of the intestinal function might have an advantage against other biomarkers. In addition to the fact that the monitoring system is already established, the amount of blood sample required for the assay is small, leading to a reduction in the burden on both newborns and health care workers. The Discussion section has been revised to highlight the potential advantage of using blood citrulline against other biomarkers. We thank again to the reviewer for the comment.
Reviewer comment:
- In the CONCLUSION it is possible to look forward not only from the future direction of the research, but it is recommended to add an outlook on the future clinical applications of this research.
- Response to the reviewer: Considering the observational nature of our study, we had hesitated to mention the clinical utility of the citrulline-based assessment of intestinal function in newborn infants. Nonetheless, the authors agree that the readers may like to know the future clinical application of blood citrulline-based diagnosis in newborn infants alongside with the direction of studies. We have revised the Conclusion section to highlight the issue.
Reviewer comment:
- Journal abbreviations should be italicized in references. Also, journals with partial abbreviations that are the same as the full title should not be followed by ".". For example, "Gastroenterology" on line 424.
- Response to the reviewer: The authors thank the reviewer for pointing the formatting errors in the Reference list, which were corrected in the revised version.
Reviewer comment:
Comments on the Quality of English Language
Please check the English expression of the article again, remove some unnecessary and repetitive words to make the article more concise, and also check the English tense.
- Response to the reviewer: We thank the reviewer for the suggestion. We have collaborated with a professional editor to streamline the manuscript. However, please note that our primary focus is on precision rather than the fluency of the text. This emphasis becomes particularly significant when the authors are not native English speakers. We hope the reviewer agrees to us on this point.
Reviewer 3 Report
Comments and Suggestions for Authors
the work is interesting but there are some things that in my opinion should be changed.
1. introduction. lines 28-33. reference is made to births in the UK and then to survival in Japan. I think it would be appropriate, to give relevance to the work, to include a brief survey of the situation including a greater number of countries (citing a review, for example?)
2. line 51 and following. the text seems to return to what has already been mentioned in line 45. it would be appropriate to organize the paragraphs more coherently.
3. being a mainly statistical-based work, I would be a little more technical in describing the analyses, in section 2.3. and provide some references
4. I wonder if it makes sense to repeat in the text the same data that are already reported in the tables. The large amount of numerical data confuses and masks the result which, perhaps, could instead be made more evident. What is greater/less than what....
5. section 3.3. Maybe I'm wrong, but it seems to me that the data reported do not correspond to those in the table. please check
6. section 3.4. Is it possible to separate the part about early postnatal from late postnatal with a new line? it is difficult at first glance to recover the data
7. figure 2. The lines have very similar colors. If printed in black/white they are not recognisable. please differentiate the lines and put a legend next to the figures.
8. figure 2. The lines have very similar colors. If printed in black/white they are not recognisable. Please differentiate the lines (by adding dot, squares...) and place the legend next to the figures. furthermore, the legend is poorly formatted
9. the discussion is too long and complicated. some parts would be better in the introduction; other parts are repetition of the data reported in the results. the integration between literature and new data is a bit lacking, superficial... despite numerous lines having been dedicated to it
Author Response
Comments by the reviewers and author rebuttals:
Reviewer 3
Reviewer comment:
the work is interesting but there are some things that in my opinion should be changed.
- Response to the reviewer: The authors are grateful to the reviewer for the encouraging comment and constructive criticisms. We have undergone a major revision to address the valuable comments provided by the reviewer. We hope the revised manuscript is improved in its clarity.
- introduction. lines 28-33. reference is made to births in the UK and then to survival in Japan. I think it would be appropriate, to give relevance to the work, to include a brief survey of the situation including a greater number of countries (citing a review, for example?) 
- Response to the reviewer: We accept that the issue regarding the preterm birth should be expanded in the section. The Introduction section has been revised by citing an additional reference: WHO recommendations for care of the preterm or low-birth-weight infant. We appreciate the reviewer’s comment, which improved the flow of the Introduction section significantly.
- line 51 and following. the text seems to return to what has already been mentioned in line 45. it would be appropriate to organize the paragraphs more coherently.
- Response to the reviewer: The text in line 51 describes the importance of early establishment of “enteral” feeding, whereas the text in line 45 summarizes the potential benefit of “parenteral” feeding. However, redundant descriptions in other part of the text have been revised for clarity. The authors thank the reviewer for the comment.
Reviewer comment:
- being a mainly statistical-based work, I would be a little more technical in describing the analyses, in section 2.3. and provide some references
- Response to the reviewer: The reviewer is right that the analytical approach should be described in more detail considering that the structure of advanced statistical models is various. We have revised the Data analysis section with additional information. We thank the reviewer for the suggestion.
Reviewer comment:
- I wonder if it makes sense to repeat in the text the same data that are already reported in the tables. The large amount of numerical data confuses and masks the result which, perhaps, could instead be made more evident. What is greater/less than what....
- Response to the reviewer: Once again, we were sorry that the presentation of findings was relatively redundant. Values shown in Tables have now been removed except for the essential information. Regarding the regression coefficient and its confidence interval, the authors would like to keep the information within the text, because an increasing number of scientific journals now recommend to show these values (but not p-values) in the text.
Reviewer comment:
- section 3.3. Maybe I'm wrong, but it seems to me that the data reported do not correspond to those in the table. please check
- Response to the reviewer: The regression coefficient and its confidence interval presented in the Results section are of interactions, such as Gestational age x Postnatal age and NEC x Postnatal age. We have double-checked the values and confirmed that they were consistent between the text and Table.
Reviewer comment:
- section 3.4. Is it possible to separate the part about early postnatal from late postnatal with a new line? it is difficult at first glance to recover the data
- Response to the reviewer: We understand that, without a new line, findings from Early and Late samples could be confused. We have inserted a line change in the revised Results section. We thank the reviewer for the suggestion.
Reviewer comment:
- figure 2. The lines have very similar colors. If printed in black/white they are not recognisable. please differentiate the lines and put a legend next to the figures.
- figure 2. The lines have very similar colors. If printed in black/white they are not recognisable. Please differentiate the lines (by adding dot, squares...) and place the legend next to the figures. furthermore, the legend is poorly formatted
- Response to the reviewer: The authors are sorry that the legend for the figure had been merged in the text in the Results section of the original manuscript. We have revised the manuscript and figure. We are grateful to the reviewer for the suggestion.
Reviewer comment:
- the discussion is too long and complicated. some parts would be better in the introduction; other parts are repetition of the data reported in the results. the integration between literature and new data is a bit lacking, superficial... despite numerous lines having been dedicated to it
- Response to the reviewer: The authors thank the reviewer for the comment. Although we had tried our best to discuss important issues within a limited space, we understand that several paragraphs were redundant in part because of our linguistic ability. We have worked on the Discussion section with a native editor to make it concise.
Round 2
Reviewer 3 Report
Comments and Suggestions for Authors
The authors have addressed and solved the main issues proposed in my previous comment. I believe the manuscript is ready for publication.